# LEARNING AWARENESS MODELS

**Brandon Amos**[1][*] **Laurent Dinh**[2] **Serkan Cabi**[3] **Thomas Rothörl**[3] **Sergio Gómez Colmenarejo**[3]
**Alistair Muldal**[3] **Tom Erez**[3] **Yuval Tassa**[3] **Nando de Freitas**[3,4] **Misha Denil**[3]

[1]Carnegie Mellon University  [2]University of Montreal  [3]DeepMind  [4]CIFAR

## ABSTRACT

We consider the setting of an agent with a fixed body interacting with an unknown and uncertain external world. We show that models trained to predict proprioceptive information about the agent's body come to represent objects in the external world. In spite of being trained with only internally available signals, these dynamic body models come to represent external objects through the necessity of predicting their effects on the agent's own body. That is, the model learns holistic persistent representations of objects in the world, even though the only training signals are body signals. Our dynamics model is able to successfully predict distributions over 132 sensor readings over 100 steps into the future and we demonstrate that even when the body is no longer in contact with an object, the latent variables of the dynamics model continue to represent its shape. We show that active data collection by maximizing the entropy of predictions about the body—touch sensors, proprioception and vestibular information—leads to learning of dynamic models that show superior performance when used for control. We also collect data from a real robotic hand and show that the same models can be used to answer questions about properties of objects in the real world. Videos with qualitative results of our models are available at `https://goo.gl/mZuqAV`.

## 1 INTRODUCTION

> *Situation awareness is the perception of the elements in the environment within a volume of time and space, and the comprehension of their meaning, and the projection of their status in the near future.* — Endsley (1987)

As artificial intelligence moves off of the server and out into the world at large; be this the virtual world, in the form of simulated walkers, climbers and other creatures (Heess et al., 2017), or the real world in the form of virtual assistants, self driving vehicles (Bojarski et al., 2016), and household robots (Jain et al., 2013); we are increasingly faced with the need to build systems that understand and reason about the world around them.

When building systems like this it is natural to think of the physical world as breaking into two parts. The first part is the platform, the part we design and build, and therefore know quite a lot about; and the second part is everything else, which comprises all the strange and exciting situations that the platform might encounter. As designers, we have very little control over the external part of the world, and the variety of situations that might arise are too numerous to anticipate in advance. Additionally, while the state of the platform is readily accessible (e.g. through deployment of integrated sensors), the state of the external world is generally not available to the system.

The platform hosts any sensors and actuators that are part of the system, and importantly it can be relied on to be the same across the wide variety situations where the system might be deployed. A virtual assistant can rely on having access to the camera and microphone on your smart phone, and the control system for a self driving car can assume it is controlling a specific make and model of vehicle, and that it has access to any specialized hardware installed by the manufacturer. These consistency assumptions hold regardless of what is happening in the external world.

This same partitioning of the world occurs naturally for living creatures as well. As a human being your platform is your body; it maintains a constant size and shape throughout your life (or at least

---

[*]Work done while BA and LD were interns at DeepMind.

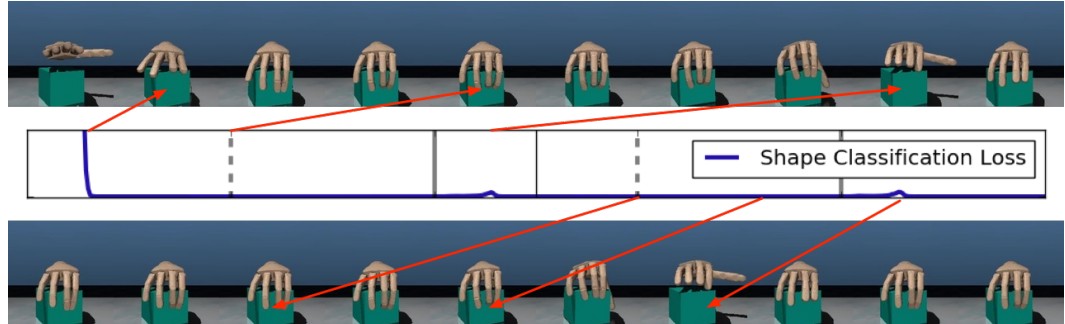

Figure 1: Illustration of a preprogrammed grasp and release cycle of a single episode of the MPL hand. The target block is only perceivable to the agent through the constraints it imposes on the movement of the hand. Note that the shape of the object is correctly predicted even when the hand is not in contact with it. That is, the hand neural network sensory model has learned persistent representations of the external world, which enable it to be aware of object properties even when not touching the objects.

these change vastly slower than the world around you), and you can hopefully rely on the fact that no matter what demands tomorrow might make of you, you will face them with the same number of fingers and toes.

This story of partitioning the world into the self and the other, that exchange information through the body, suggests an approach to building models for reasoning about the world. If the body is a consistent vehicle through which an agent interacts with the world and proprioceptive and tactile senses live at the boundary of the body, then predictive models of these senses should result in models that represent external objects, in order to accurately predict their future effects on the body. This is the approach we take in this paper.

We consider two robotic hand bodies, one in simulation and one in reality. The hands are induced to grasp a variety of target objects (see Figure 1 for an example) and we build forward models of their proprioceptive signals. The target objects are perceivable only through the constraints they place on the movement of the body, and we show that this information is sufficient for the dynamics models to form holistic, persistent representations of the targets. We also show that we can use the learned dynamics models for planning, and that we can illicit behaviors from the planner that depend on external objects, in spite of those objects not being included in the observations directly (see Figure 7).

Our simulated body is a model of the hand of the Johns Hopkins Modular Prosthetic Limb (Johannes et al., 2011), realized in MuJoCo (Todorov et al., 2012). The model is actuated by 13 motors each capable of exerting a bidirectional force on a single joint. The model is also instrumented with a series of sensors measuring angles and torques of the joints, as well as pressure sensors measuring contact forces at several locations across its surface. There are also inertial measurement units located at the end of each finger which measure translational and rotational accelerations. In total there are 132 sensor measurements whose values we predict using our dynamics model.

Our real body is the Shadow Dexterous Hand, which is a real robotic hand with 20 degree of freedom control. This allows us to show that that our ideas apply not only in simulation, but succeed in the real world as well. The Shadow Hand is instrumented with sensors measuring the tension of the tendons driving the fingers, and also has pressure sensors on the pad of each fingertip that measure contact forces with objects in the world. We apply the same techniques used on the simulated model to data collected from this real platform and use the resulting model to make predictions about states of external objects in the real world.

## 2 RELATED WORK

**Intrinsic motivation and exploration:** Given our goal to gather information about the world and, and in particular to actively seek out information about external objects, our work is naturally related

to work on intrinsic motivation. The literature on intrinsic motivation is vast and rich, and we do not attempt to review it fully here. Some representative works include Oudeyer & Kaplan (2008; 2009); Sequeira et al. (2011); Still & Precup (2012); Bellemare et al. (2016); Martius et al. (2013); Schmidhuber (2008); Mohamed & Rezende (2015); Haber et al. (2018b;a) Some of the ideas here, in particular the notion of choosing actions specifically to improve a model of the world, echo earlier speculative work of Schmidhuber (1991) and Storck et al. (1995).

Several authors have implemented intrinsic motivation, or curiosity based objectives in visual space, through predicting interactions with objects (Pinto et al., 2016), or through predicting summary statistics of the future (Venkatraman et al., 2017; Downey et al., 2017). Other authors have also investigated using learned future predictions directly for control (Dosovitskiy & Koltun, 2016).

Many works formulate intrinsic motivation as a problem of learning to induce errors in a forward model, possibly regularized by an additional inverse model (Pathak et al., 2017; de Abril & Kanai, 2018). However, since we use planning, rather than policies, for active control we cannot adapt their objectives directly. Objectives that depend on the observed error in a prediction cannot be rolled forward in time, and thus we are forced to work with similar, but different objectives in our planner.

When stochastic transition and observation models are available, it is possible to use simulation to infer optimal plans for exploring environments (Martinez-Cantin et al., 2009). Our setting uses predominantly deterministic distributed representations, and our models are learned from data.

A sea of other methods have been proposed for exploration (MacKay, 1992; Ghavamzadeh et al., 2015; Asmuth et al., 2009; Gal, 2016; Stachniss et al., 2005; Plappert et al., 2017; Fu et al., 2017). Our approach builds on this literature.

**Haptics:** Humans use their hands to gather information in structured task driven ways (Lederman & Klatzky, 1987); and it will become clear from the experiments why hands are relevant to our work. Our interest in hands and touch brings us into contact with a vast literature on haptics (Zheng et al., 2016; Gao et al., 2016; Cao et al., 2016; Loeb, 2013; Edmonds et al., 2017; Su et al., 2015; Navarro et al., 2012; Aggarwal et al., 2015; Liu et al.; Sung et al., 2017; Ciobanu et al., 2013; Karl et al., 2016; Su et al., 2012).

There is also work in robotics on using the anticipation of sensation to guide actions (Indranil Sur, 2017), and on showing how touch sensing can improve the performance of grasping tasks (Calandra et al., 2017). Model based planning has been very successful in these domains (Deisenroth & Rasmussen, 2011).

**Sequence-to-sequence modelling:** There has been a lot of recent interest in sequence-to-sequence modelling (Downey et al., 2017; Venkatraman et al., 2017; Chung et al., 2015; Fraccaro et al., 2016; Bayer & Osendorfer, 2014; Archer et al., 2015; Krishnan et al., 2015), particularly in the context of predicting distributions and in dynamics modelling in reinforcement learning. In this paper we use a sequence to sequence variant that shares weights between the encoder and decoder portions of the model.

**Predicting unknown quantities in RL:** The consciousness prior (Bengio, 2017) considers recurrent latent dynamics models similar to ours and suggests mapping from their hidden states to other spaces that aren't directly modelled.

Yu et al. (2017) propose a method of learning control policies that operate under unknown dynamics models. They consider the dynamics model parameters as an unobserved part of the state, and train a system identification model to predict these parameters from a short history of observations. The predictions of the system identification model are used to augment the observations which are then fed to a *universal policy*, which has been trained to act optimally under an ensemble of dynamics models, when the dynamics parameters are observed. The key contribution of their work is a training procedure that makes this two stage modelling process robust.

Although the high level motivation of Yu et al. (2017) is similar, there is an obvious analogy between their system identification model and our diagnostics, many of the specifics are quite different from the work presented here. They explicitly do not consider memory-based tasks (the system identification model looks only at a short window of the past) whereas one of our key interests is in how our models preserve information in time. They also use a two stage training process, where both stages of training require knowledge of the system parameters; it is only at test time where these are

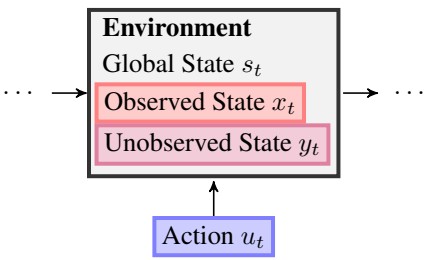

**Definitions**

**Dynamics Model:** Predicts the action-conditional future observations given the past observations and actions: $p(x_{t+1:t+k}|u_{1:t+k-1}, x_{1:t})$

**Awareness:** The information about unobserved states that is represented by the dynamics model.

**Diagnostic Model:** A model used to evaluate (or diagnose) the awareness of a dynamics model by predicting unobserved states.

Figure 2: Overview of our notation and definitions.

unknown. In contrast, we use the system parameters only as an analysis strategy, and at no point require knowledge of them to train the system. Finally, the bodies we consider (robot hands) are substantially more complex than those of Yu et al. (2017), and we do not make use of an explicit parameterization of the system dynamics.

The work of Fu et al. (2016) also fits dynamics models using neural networks and uses planning in these models to guide action selection. They train a global dynamics model on data from several tasks, and use this global model as a prior for fitting a much simpler local dynamics model within each episode. The global model captures course grained dynamics of the robot and its environment, while the local model accounts for the specific configuration of the environment within an episode. Although they do not probe for this explicitly, one might hypothesize that the type of awareness of the environment that we are after in this work could be encoded in the parameters of their local models.

## 3 Dynamics, Awareness, and Diagnostics

We consider an agent operating in a discrete-time setting where there is a stochastic *unobservable global state* $s_t \in \mathcal{S}$ at each timestep $t$ and the agent obtains a stochastic *observation* $x_t \in \mathcal{X}$ where $\mathcal{X} \subseteq \mathcal{S}$ and takes some *action* $u_t \in \mathcal{U}$. Our goal is to learn a predictive *dynamics model* of the agent's action-conditional future observations $p(x_{t+1:t+k}|u_{1:t+k-1}, x_{1:t})$ for $k$ timesteps into the future given all of the previous observations and actions it has taken. We assume that the dynamics model has some hidden state it uses to encode information in the observed trajectory. We will then use these models to reason about the global state $s_t$ even though no information about this state is available during training, which we refer to as *awareness*. Figure 2 summarizes the notation and definitions we use throughout the rest of the paper.

To show that information required for reasoning is present in the states of our dynamics models we use auxiliary models, which we call *diagnostic models*. A diagnostic model looks at the states of a dynamics model and uses them to to predict an interpretable *unobserved state* in the world $y_t \in \mathcal{Y}$, where $\mathcal{Y} \subseteq \mathcal{S}$ and in most cases $\mathcal{X} \cap \mathcal{Y} = \emptyset$. When training a diagnostic model we allow ourselves to use privileged information to define the loss, but we do not allow the diagnostic loss to influence the representations of the dynamics model.

The diagnostic models are a post-hoc analysis strategy. The dynamics models are trained using only the observed states, and then frozen. After the dynamics models are trained we train diagnostic models on their states, and the claim is that if we can successfully predict properties of unobserved states using diagnostic models trained in this way then information about the external objects is available in the states of the dynamics model.

## 4 The Predictor-Corrector (PreCo) Dynamics Model

This section introduces the Predictor-Corrector (PreCo) dynamics model we use for long-horizon multi-step predictions over the observation space $p(x_{t+1:t+k}|u_{1:t+k-1}, x_{1:t})$. We first encode the observed trajectory $\{u_{1:t}, x_{1:t}\}$ into a *deterministic* hidden state $h_t \in \mathcal{H}$ using a recurrent model

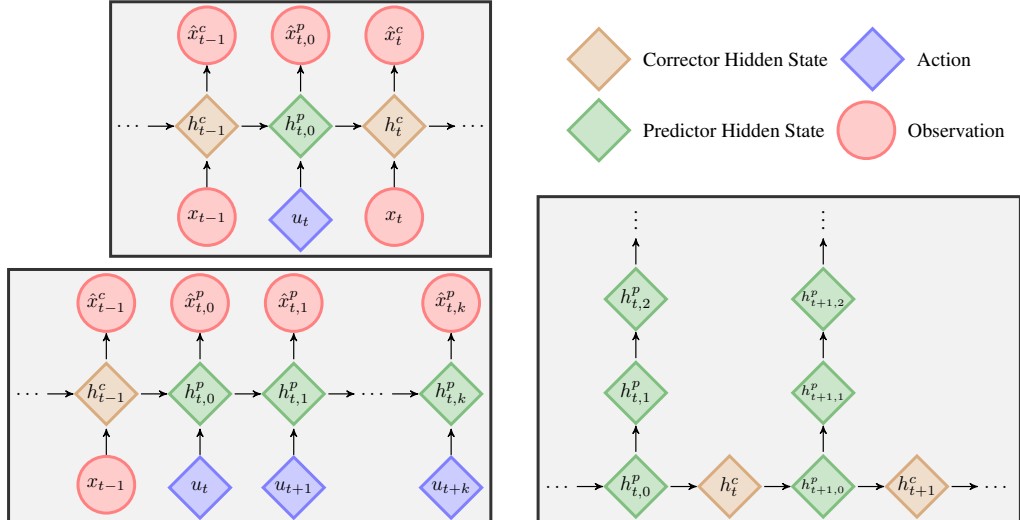

Figure 3: **Top Left:** A PreCo model generating single-step predictions and corrections, as in optimal filtering. **Bottom Left:** A PreCo model making multi-step predictions. **Right:** Multi-step rollouts, from all timesteps, are used for fitting a PreCo model to a trajectory. Deterministic nodes are represented with diamonds and stochastic nodes are represented with circles.

parameterized by $\theta$ and then use this hidden state to predict distributions over the future observations $x_{t+1:t+k}$. We show experimentally that even though the hidden states $h_t$ were only trained on *observed states*, they contain an *awareness* of *unobserved states* in the environment.

Using a deterministic hidden state allows us to easily unroll the predictor without needing to approximate the distributions with sampling or other approximate methods. We assume that the observation predictions are independent of each other given the hidden state, and can be modeled as

$$p(x_{t+1:t+k}|u_{t:t+k-1}, h_{t:t+k}) = \prod_{\kappa=1}^{k} p(x_{t+\kappa}|u_{t:t+\kappa-1}, h_{t:t+\kappa})$$

This modelling is done with three deterministic components:

1. $\text{Predictor}_\theta : \mathcal{H} \times \mathcal{U} \rightarrow \mathcal{H}$ predicts the next hidden state after taking an action,

2. $\text{Corrector}_\theta : \mathcal{H} \times \mathcal{X} \rightarrow \mathcal{H}$ corrects the current hidden state after receiving an observation from the environment, and

3. $\text{Decoder}_\theta : \mathcal{H} \rightarrow P_\mathcal{X}$ maps from the hidden state to a distribution over the observations.

Separating the dynamics model into predictor and corrector components allows us to operate in single-step and multi-step prediction modes as Figure 3 shows. The predictor can make action-conditional predictions using the hidden states from the corrector for single-step predictions as $h_{t,0}^p = \text{Predictor}_\theta(h_{t-1}^c, u_t)$ or from itself for multi-step predictions as $h_{t,i+1}^p = \text{Predictor}_\theta(h_{t,i}^p, u_{t+i})$. In our notation, $h_{t,i}^p$ denotes the predictor's hidden state prediction at time $t + i$ starting from the corrector's state at time $t - 1$. The corrector then makes the updates $h_t^c = \text{Corrector}_\theta(h_{t,0}^p, x_t)$.

To train PreCo models, we maximize the likelihood on a reference set of trajectories using the single-step predictions as well as multi-step predictions stemming from *every* timestep. The structure of the resulting graph of only the hidden states is shown on the right of Figure 3, omitting the observed states, trajectories, and predicted distributions. We call this technique *overshooting*, and we call the number of steps predicted forward by the decoder the overshooting length.

The predictor and corrector components use single layer LSTM cores. We embed the inputs with a separate embedding MLP for the controls and sensors. We predict independent mixtures of Gaus-

sians at every step with a mixture density network (Bishop, 1994). Each dimension of each prediction is an independent mixture. We use separate MLPs to produce the means, standard deviations and mixture weights. We use Adam (Kingma & Ba, 2014) for parameter optimization.

## 5 CONTROL WITH DYNAMICS AND DIAGNOSTIC MODELS

Model predictive control (MPC), the strategy of controlling a system by repeatedly solving a model-based optimization problem in a receding horizon fashion, is a powerful control technique when a dynamics model is known. Throughout this paper, we use MPC to achieve objectives based on predictions from our dynamics models. Formally, MPC requires that at each timestep after receiving an observation and correcting the hidden state, we solve the optimization problem

$$
\begin{aligned}
h^\star_{1:T}, u^\star_{1:T} \;=\; &\operatorname*{argmin}_{h_{1:T}, u_{1:T}} \; \sum_t C(h_t, u_t) \\
&\text{subject to} \;\; h_0 = h_{\text{init}} \\
&\qquad\qquad\quad h_{t+1} = \text{Predictor}_\theta(h_t, u_t) \\
&\qquad\qquad\quad u_{1:T} \in \mathcal{U}_{1:T}
\end{aligned}
\tag{1}
$$

where the timesteps in this problem are offset from the actual timestep in the real system, the initial hidden state $h_{\text{init}}$ is from the most recent corrector's state, and the remaining hidden states are unrolled from the predictor. In our experiments we also add constraints to the actions $\mathcal{U}_{1:T}$ so that they lie in a box $||u_t||_\infty \leq 1$ and we enforce slew rate constraints $||u_{t+1} - u_t||_\infty \leq 0.1$. After solving this problem, we execute the first returned control $u^\star_1$ on the real system, step forward in time, and repeat the process.

This formulation allows us to express standard objectives defined over the observation space by using the decoder to map from the hidden state to a distribution over observations at each timestep. We can also use other learned models, such as diagnostic models, to map from the hidden state to other unobservable quantities in the world.

Our MPC solver for (1) uses a shooting method with a modified version of Adam (Kingma & Ba, 2014) to iteratively find an optimal control sequence from some initial hidden state. At every iteration, we unroll the predictor, compute the objective at each timestep, and use automatic differentiation to compute the gradient of the objective with respect to the control sequence. To handle control constraints, we project onto a feasible set after each Adam iteration. During an episode, we warm-start the nominal control and hidden state sequence to the appropriately time-shifted control and hidden state sequence from the previous optimal solution.

## 6 COLLECTING TRAJECTORIES AND EXPLORATION

Learning dynamics models such as the PreCo model in Section 4 requires a collection of trajectories. In this section, we discuss two ways of collecting trajectories for training dynamics models: *passive* collection does not use any input from the dynamics model while *active* collection seeks to actively improve the dynamics model.

### 6.1 PASSIVE COLLECTION

The simplest data collection strategy is to hand design a behavior that is independent of the dynamics model. Such strategies can be completely open loop, for example taking random actions driven by a noise process, and also encompass closed loop policies such as following a pre-programmed nominal trajectory. In these situations, we are only interested in learning a dynamics model to achieve an awareness of the unobserved states, not to make policy improvements. We use the term *passive collection* to emphasize that the the data collection behavior does not depend on the model being trained. Once collected, the maximum likelihood PreCo training procedure described in Section 4 can be used to fit the PreCo dynamics model to the trajectories, but there is no feedback between the state of the dynamics model and the data collection behavior.

## 6.2 Active collection: Exploration through maximizing uncertainty

Beyond passive collection we can consider using using the dynamics model to guide exploration towards parts of the state space where the the model is poor. We call this process *active collection* to emphasize that the model being trained is also being used to guide the data collection process. This section describes the method of active collection we use in the experiments.

In this paper, we consider environments that are entirely deterministic, except for a stochastic initial unobserved state that the observed state can gather information about. When our dynamics model over the observed state makes uncertain predictions, the source of that uncertainty stems from one of two places: (1) the model is poor, as a consequence of there being too little data or from too small capacity, or (2) properties of the external objects are not yet resolved by the observations seen so far.

Our active exploration exploits this fact by choosing actions to maximize the uncertainty in the rollout predictions. An agent using this uncertainty maximization policy attempts to seek actions for which the outcome is not yet known. This uncertainty can then be resolved by executing these actions and observing their outcome, and the resulting trajectory of observations, actions, and sensations can be used to refine the model.

To choose actions to gather information we use MPC as described in Section 5 over an objective that maximizes the uncertainty in the predictions. Our predictions are Mixtures of Gaussians at each timestep, and the uncertainty over these distributions can be expressed in many ways. We use the Rényi entropy of our model predictions as our measure of uncertainty because it can be easily computed in closed form. Concretely, for a single Mixture of Gaussians prediction $f(x)$ we can write

$$H_2(f) = -\log\left[\int f(x)^2\,\mathrm{d}x\right] = -\log\left[\sum_{ij}\alpha_i\alpha_j\frac{\exp\left\{-\frac{(\mu_i-\mu_j)^2}{2(\sigma_i^2+\sigma_j^2)}\right\}}{\sqrt{2\pi}\sqrt{\sigma_i^2+\sigma_j^2}}\right]$$

where $i$ and $j$ index the mixture components in the likelihood. A more complete derivation is shown in Appendix A, which extends the result of Wang et al. (2009) to the case when the mixture components have different variances. We obtain an information seeking objective by summing the entropy of the predictions across observations and across time, which is expressed as the cost function in MPC (1) as $C(h_t, u_t) = -\sum_{f_i} H_2(f_i)$ where, through a slight abuse of notation, $f_i \in \mathrm{Decoder}_\theta(h_t)$ is a distribution over the observation dimension $i$.

We implement this information gathering policy to collect training data for the model in which it is planning. In our implementation these are two processes running in parallel: we have several actors each with a copy of the current model weights. These use MPC to plan and execute a trajectory of actions that maximizes the model's predicted uncertainty over a fixed horizon trajectory into the future. The observations and actions generated by the actors are collected into a large shared buffer and stored for the learner.

While the actors are collecting data, a single learner process samples batches of the collected trajectories from the buffer being written to by the actors. The learner trains the PreCo model by maximum likelihood as described in Section 4, and the updated model propagates back to the actors who continue to plan using the updated model. We implemented this using the framework of Horgan et al. (2018).

## 7 Experiments on the Simulated MPL Hand

### 7.1 The MPL hand environment

Our simulated environment consists of a hand with a random object placed underneath of it in each episode. The observation state space consists of sensor readings from the hand, and the unobserved state space consists of properties of the object.

The hand is from the Johns Hopkins Modular Prosthetic Limb (Johannes et al., 2011) which we refer to as the "MPL hand", or simply "the hand". This model is distributed with the MuJoCo HAPTIX

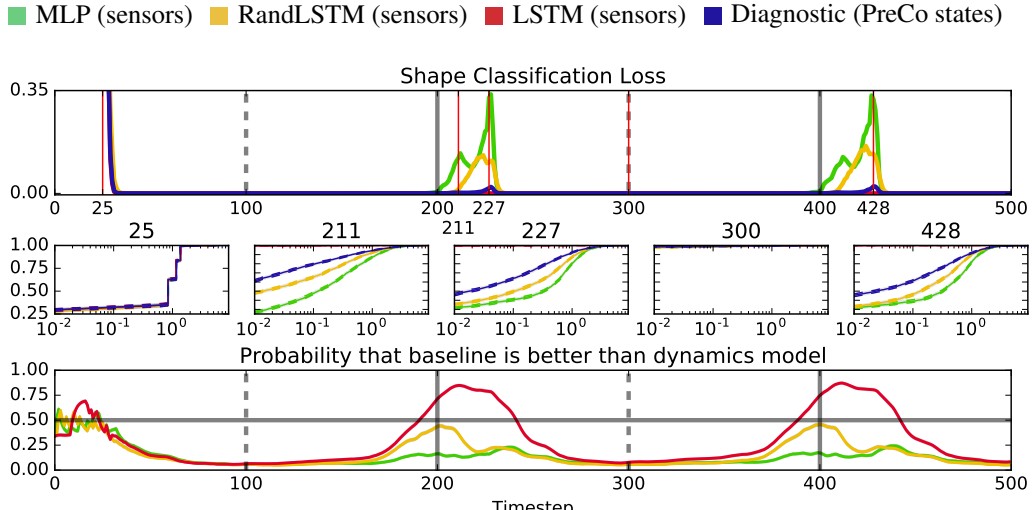

Figure 4: Results for the passive data collection experiment. Black vertical lines mark timesteps where the hand is fully open (solid) or fully closed (dashed). See the main text for a description of the different baseline models. The baseline models are end-to-end supervised on the shape classification task, whereas the PreCo states are learned without the shape information. **Top:** Classification loss vs episode timestep for the diagnostic model and the three baselines. Lines show median loss averaged over 5000 test episodes. **Middle:** Cumulative distributions of loss at the indicated timesteps. **Bottom:** Curves showing the median probability that each baseline model achieves lower classification loss than the PreCo diagnostic model, as a function of episode timesteps. Computed by directly comparing loss values.

software and is available for download from the MuJoCo website.[1] The hand is actuated by 13 motors and has sensors that provide a 132 dimensional observation, which we describe in more detail in Appendix C.

In each episode the hand starts suspended above the table with its palm facing downwards. A random geometric object that we call the "target" is placed on the table, and the hand is free to move to grasp or manipulate the object. The shape of the target is randomly chosen in each episode to be a box, cylinder or ellipsoid and the size and orientation of the target are randomly chosen from reasonable ranges. Figures 1 and 7 show renderings of the environment.

### 7.2 AWARENESS THROUGH PASSIVE DATA COLLECTION

We begin by exploring awareness in the passive setting, as a pure supervised learning problem. We manually design a policy for the hand, which executes a simple grasping motion that closes the hand about the target it and then releases it. We generate data from the environment by running this grasp-and-release cycle three times for each episode. Using the dataset generated by the grasping policy, we train a PreCo model described in Section 4. The full set of hyperparameters for this model can be found in Appendix D.

We evaluate the awareness of our model by measuring our ability to predict the shape of the target at each timestep. We are especially interested in the predictions at timesteps where the hand is not in direct contact with the target, since these are the points that allow us to measure the persistence of information in the dynamics model. We expect that even a naïve model should have enough information to identify the target shape at the peak of a grasp, but our model should do a better job of preserving that information once contact has been lost.

Recall from Section 3 that the diagnostic model we use to predict the target shape is trained in a second phase after the dynamics model is fully trained. The identity of the target shape is used when training the diagnostic model, but is not available when training the dynamics model, and the

---

[1]http://www.mujoco.org/book/haptix.html

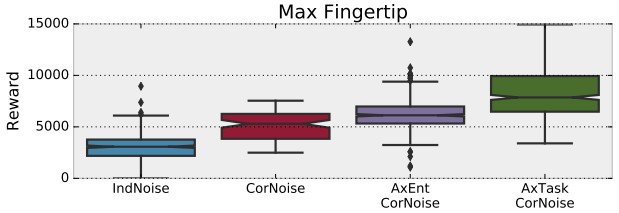 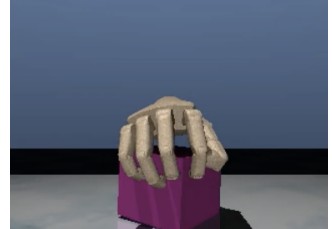

Figure 5: **Left:** Reward obtained by planning to achieve the max fingertip objective using dynamics models trained with different data collection strategies. **Right:** A frame from a planned max fingertip trajectory.

training of the diagnostic does not modify the learned dynamics model, meaning that no information from the diagnostic loss is able to leak into the states of the dynamics model.

Figure 4 shows the results of this experiment. We compare the diagnostic predictions trained on the features of our dynamics model to three different baselines that do not use the dynamics model features.

1. The **MLP** baseline uses an MLP trained to directly classify the target from the sensor readings of the hand, ignoring all dependencies between timesteps within an episode. We expect this baseline to give a lower bound on performance of the diagnostic. This is an important baseline to have since as the hand opens there is still residual information about the shape of the target in the position of joints, which is identified by this baseline.

2. The **LSTM** baseline uses an LSTM trained to directly classify the target from the sensor readings of the hand, taking full account of dependencies between timesteps within an episode. Since this baseline has access to the target class at training time, and is also able to take advantage of the temporal dependence within each episode, we expect it to give an upper bound on performance of the diagnostic model, which only has access to the states of the pre-trained PreCo model.

3. The **RandLSTM** baseline finds a middle ground between the MLP and LSTM models. We use the same architecture and training procedure as for the LSTM baseline, but we do not train the input or recurrent weights of the model. By comparing the performance of this baseline to the diagnostic model we can see that our success cannot be attributed merely to the existence of an arbitrary temporal dependence.

The results in Figure 4 show that the dynamics PreCo model reliably preserves information about the identity of the target over time, even though this information is not available to the model directly either in the input or in the training loss.

## 7.3 AWARENESS THROUGH ACTIVE DATA COLLECTION

In this section we explore how different data collection strategies lead to models of different quality. We evaluate the quality of the trained models by using them in an MPC planner to execute a simple diagnostic control task.

The diagnostic task we use in this section is to maximize the total pressure on the fingertip sensors on the hand. This is a good task to evaluate these models because the most straightforward way to maximize pressure on the fingers is to squeeze the target block, and demonstrating that we can achieve this objective through MPC shows that the models are able to anticipate the presence of the block, and reason about its effect on the body.

Note that because we act through planning, the implication for the representations of the model are stronger than they would be if we trained a policy to achieve the same objective. A policy need only learn that when the hand is open it should be closed to reach reward. In contrast, a model must learn that closing the hand will lead the fingers to encounter contacts, and it is only later that this prediction is turned into a reward for evaluation.

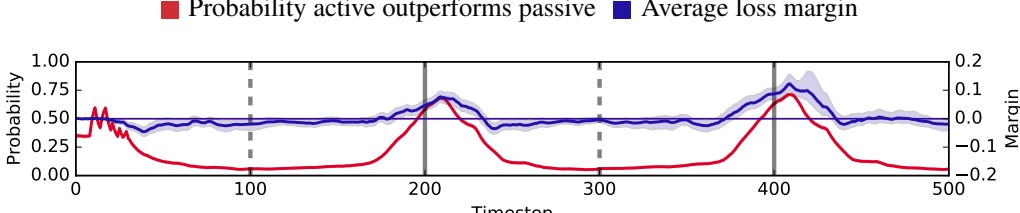

Figure 6: Diagnostic comparison between active and passive data collection computed by directly comparing loss values. The model trained with actively collected data outperforms its passive counterpart in regions of the grasp trajectory where the hand is not in contact with the block.

We compare several different data collection policies, and their performances on the diagnostic task are shown in Figure 5.

1. The **IndNoise** policy collects data by executing random actions sampled from a Normal distribution with standard deviation of 0.2.

2. The **CorNoise** policy collects data by executing random actions sampled from a Ornstein Uhlenbeck process with damping of 0.2, driven by an independent normal noise source with standard deviation 0.2. Each of the 13 actions is sampled from an independent process, with correlation happening only over time.

3. The **AxEnt** policy uses the MPC planner described in Section 5 to maximize the total entropy of the model predictions over a horizon of 100 steps. The objective for the planner is to maximize the Rényi entropy, as described in Section 6.2.

4. The **AxTask** policy also uses the MPC planner of Section 5 to collect data, but here the planning objective for data collection is the same as for evaluation.

For both of the planning policies we found that adding correlated noise (using the same parameters as the CorNoise policy) to the actions chosen by the planner lead to much better models. Without this source of noise the planners do not generate enough variety in the episodes and the models underperform.

We evaluate each model by running several episodes where we plan to achieve maximum fingertip pressure, and show the resulting rewards in Figure 5. Note that the evaluation objective is different than the training objective for all models except AxTask. We do not add additional noise to planned actions when running the evaluation.

We also evaluate the awareness of the AxEnt model using the shape diagnostic task from Section 7.2. Figure 6 compares the performance of a diagnostic trained on the AxEnt dynamics model to the passive awareness diagnostic of Section 7.2. The model trained with actively collected data outperforms its passive counterpart in regions of the grasp trajectory where the hand is not in contact with the block.

### 7.4 QUALITATIVE EVALUATION

In this section we present qualitative results of using a the AxEnt model to execute different objectives through planning. We do this with MPC as described in Section 5.

1. **Maximizing entropy of the predictions,** as we did during training, leads to exploratory behavior. In Figure 7 we show a typical frame from an entropy maximizing trajectory, as well as typical frames from controlling for two different objectives.

2. **Optimizing for fingertip pressure** tends to lead to grasping behavior, since the easiest way to achieve pressure on the fingertips is to push them against the target block. There is an alternative solution which is often found where the hand makes a tight fist, pushing its fingertips into its own palm. This is the same as the diagnostic task used in the previous section.

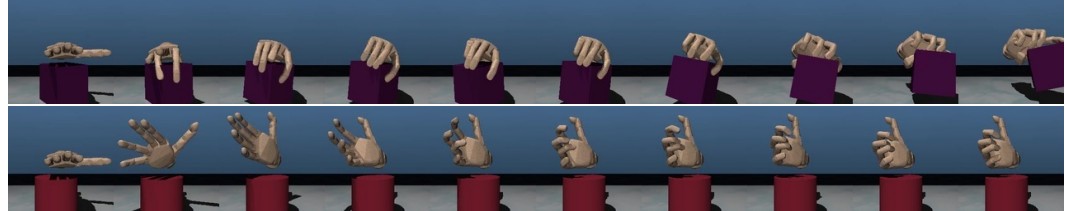

Figure 7: Examples of the hand behaving to maximize uncertainty about the future (top) or minimize uncertainty (bottom). When the hand is trained to maximize uncertainty it engages in playful behavior with the object. The body models learned with this objective, can then be re-used with novel objectives, such as minimizing uncertainty. When doing so, we see that the hand avoids contact so as to minimize uncertainty about future proprioceptive and haptic predictions.

3. **Minimizing entropy of the predictions** is also quite interesting. This is the negation of the information gathering objective, and it attempts to make future observations as uninformative as possible. Optimizing for this objective results in behavior where the hand consistently pulls away from the target object.

Qualitative results from executing each of the above policies are shown in Figures 5 and 7. The behavior when minimizing entropy of the predictions is particularly relevant. The resulting behavior causes the hand to pull away from the target object, demonstrating that the model is aware not only of how to interact with the target, but also how to avoid doing so. Videos of the model in action are available online at `https://goo.gl/mZuqAV`.

## 8 EXPERIMENTS IN THE REAL WORLD

We have shown that our models work well in simulation. We now turn to demonstrating that they are effective in reality as well.

### 8.1 THE SHADOW HAND ENVIRONMENT

We use the 24-joint Shadow Dexterous Hand[2] with 20-DOF tendon position control and set up a real life analog of our simulated environment, as shown in Figure 8. Since varying the spatial extents of an object in real life would be very labor intensive we instead use a single object fixed to a turntable that can rotate to any one of 255 orientations, and our diagnostic task in this environment is to recover the orientation of the grasped object.

We built a turntable mechanism for orienting the object beneath the hand, and design some randomized grasp trajectories for the hand to close around the block. The object is a soft foam wedge (the shape is chosen to have an unambiguous orientation) and fixed to the turntable. At each episode we turn the table to a randomly chosen orientation and execute two grasp release cycles with the hand robot.

### 8.2 DATA COLLECTION

Over the course of two days we collected 1140 grasp trajectories in three sessions of 47, 393 and 700 trajectories. We use the 47 trajectories from the initial session as test data, and use the remaining 1093 trajectories for training. Each trajectory is 81 frames long and consists of two grasp-release cycles with the target object at a fixed orientation. At each timestep we measure four different proprioceptive features from the robot:

1. The **actions**, a set of 20 desired joint positions, sent to the robot for the current timestep.
2. The **angles**, a set of 24 measured joint positions, reported by the robot at the current timestep. There are more angles than actions because not all joints of the hand are sep-

---

[2]`https://www.shadowrobot.com/products/dexterous-hand/`

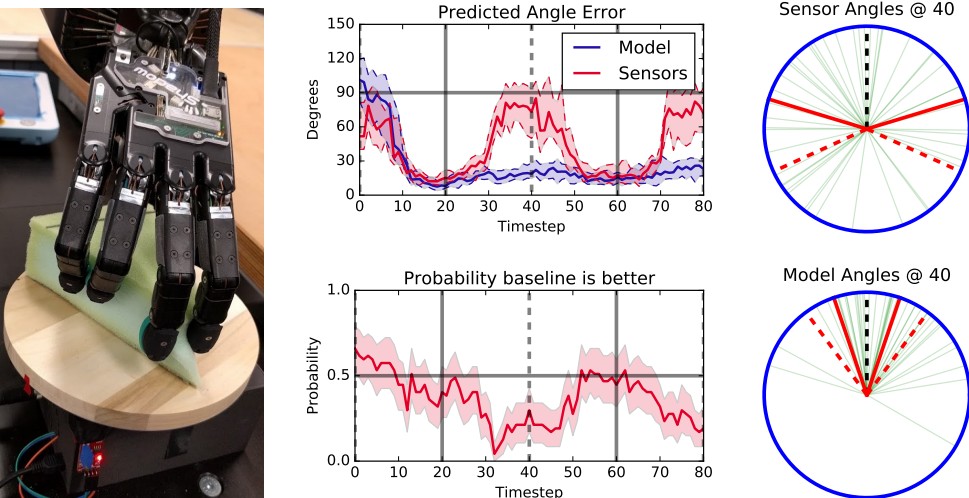

Figure 8: **Left:** The robotic hand setup. **Center:** Results on predicting block orientation with sensor data recorded from the shadow hand. The upper plot shows the median error as a function of time and the bottom plot shows a bootstrap estimate of the probability that using the model features fails to improve on using sensor measurements directly. Error regions in both plots show 95% confidence intervals, estimated by bootstrap sampling. **Right:** Predicted angles on test trajectories at step 40 using only sensor readings (top) and model features (bottom). Green lines show predicted angles for individual samples (rotated so ground truth is vertical). The solid and dashed red lines show 50 and 75 percentile error cones, respectively.

arately actuated, and the measured angles may not match the intended actions due to force limits imposed by the low level controller.

3. The **efforts**, which provide 20 distinct torque readings. Each effort measurement is the signed difference in tension between tendons on the inside and outside of one of the actuated joints.

4. The **pressures** are five scalar measurements that indicate the pressure experienced by the pads on the end of each finger.

Joint ranges of the hand are limited to prevent fingers pushing each other, and the actuator strengths are limited for the safety of the robot and the apparatus. At each grasp-release cycle final grasped and released positions are sampled from handcrafted distributions. Position targets sent to the robot are calculated by interpolating between these two positions in 20 steps.

There are multiple complexities the sensor model needs to deal with. First of all once a finger touches the object actual positions and target positions do not match, and the foam object bends and deforms. Also the hand can occasionally overcome the resistance in the turntable motor causing the target object to rotate during the episode (for about 10-20 degrees and rarely more). This creates extra unrecorded source of error in the data.

### 8.3    AWARENESS AND DIAGNOSTICS

We train a forward model on the collected data, and then treat prediction of the orientation of the block as a diagnostic task. Figure 8 shows that we can successfully predict the orientation of the block from the dynamics model state.

# 9 CONCLUSION

In this paper we showed that learning a forward predictive model of proprioception we obtain models that can be used to answer questions and reason about objects in the external world. We demonstrated this in simulation with a series of diagnostic tasks where we use the model features to identify properties of external objects, and also with a control task where we show that we can plan in the model to achieve objectives that were not seen during training.

We also showed that the same principles we applied to our simulated models are also successful in reality. We collected data from a real robotic platform and used the same modelling techniques to predict the orientation of a grasped block.

ACKNOWLEDGMENTS

BA is supported by the National Science Foundation Graduate Research Fellowship Program under Grant No. DGE1252522. We thank Dougal Sutherland and Matthew W. Hoffman for insightful discussions.

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

# A    DERIVING THE RÉNYI ENTROPY OF A MIXTURE OF GAUSSIANS

$$H_2(f) = -\log \int f(x)^2 \mathrm{d}x$$

$$= -\log \int \left( \sum_i \alpha_i f_i(x|\mu_i, \sigma_i^2) \right)^2 \mathrm{d}x$$

$$= -\log \int \sum_i \sum_j \alpha_i \alpha_j f_i(x|\mu_i, \sigma_i^2) f_j(x|\mu_j, \sigma_j^2) \mathrm{d}x$$

$$= -\log \sum_i \sum_j \alpha_i \alpha_j \int f_i(x|\mu_i, \sigma_i^2) f_j(x|\mu_j, \sigma_j^2) \mathrm{d}x$$

$$= -\log \sum_i \sum_j \alpha_i \alpha_j \frac{\exp\left\{ -\frac{(\mu_i - \mu_j)^2}{2(\sigma_i^2 + \sigma_j^2)} \right\}}{\sqrt{2\pi}\sqrt{\sigma_i^2 + \sigma_j^2}}$$

where the last step can be computed with Mathematica, and is also given in Bromiley (2003):

```
Integrate[
 PDF[NormalDistribution[Subscript[\[Mu], i], Subscript[\[Sigma], i]],
  x]*PDF[NormalDistribution[Subscript[\[Mu], j], Subscript[\[Sigma],
    j]], x], {x, -\[Infinity], \[Infinity]},
 Assumptions -> {Subscript[\[Sigma], i] \[Element] Reals,
  Subscript[\[Sigma], j] \[Element] Reals,
  Re[Subscript[\[Sigma], i]] > 0, Re[Subscript[\[Sigma], j]] > 0}]
```

# B    EXTRA RESULTS

Figures 9 and 10 show planned trajectories and model predictions when attempting to maximize fingertip pressure and to minimize predicted entropy, respectively.

# C    MPL HAND

The MPL hand is actuated by 13 motors each capable of exerting a bidirectional force on a single degree of freedom of the hand model. Each finger is actuated by a motor that applies torque to the MCP joint, and the MCP joint of each finger is coupled by a tendon to the PIP and DIP joints of the same finger, causing a single action to flex all joints of the finger together. Abduction of the main digits (ABD) is controlled by two motors attached to the outside of the index and pinky fingers, respectively. Unlike the main digits, the thumb is fully actuated, with separate motors driving each joint. The thumb has its own abduction joint, and somewhat strangely the thumb is composed of three jointed segments (unlike a human thumb which has only two). Each segment is separately controlled for a total of four actuators controlling the thumb. Finally the hand is attached to the world by fully actuated three three degree of freedom wrist joint, for a total of 13 actuators.

The hand model includes several sensors which we use as proprioceptive information. We observe the position and velocity of each joint in the model (three joints in the wrist and four in each finger except the middle which has no abduction joint, for a total of 22 joints), as well as the position, velocity and force of each of the 13 actuators. We also record from inertial measurement units (IMUs) located in the distal segment of each of the five fingers. Each IMU records three axis rotational and translational acceleration for a total of 30 acceleration measurements. Finally there are 19 pressure sensors placed throughout the inside of the hand that measure the magnitude of contact forces. Each finger including the thumb has three touch sensors, one on each segment (recall that the thumb has three segments in this model), and the palm of the hand has four different touch

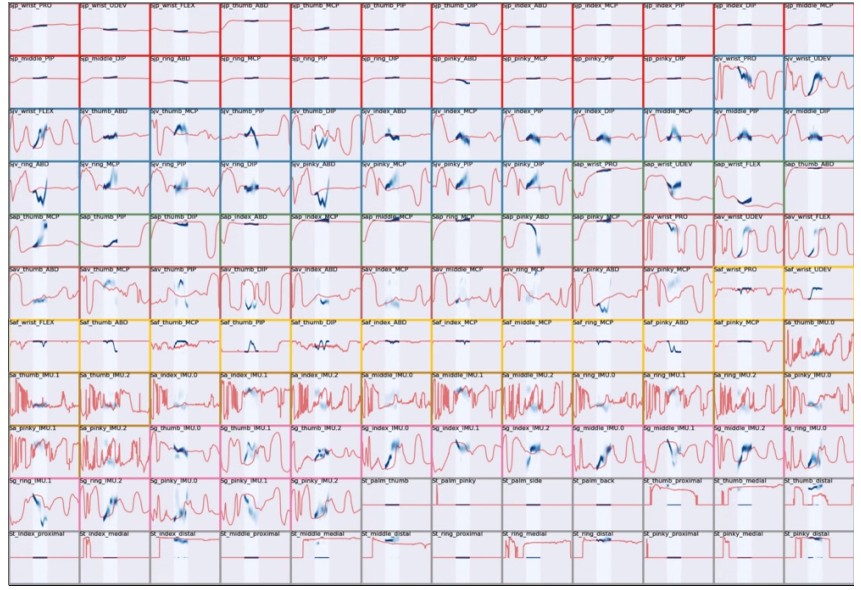

Figure 9: A visualization of the model planning to maximize predicted fingertip pressure.

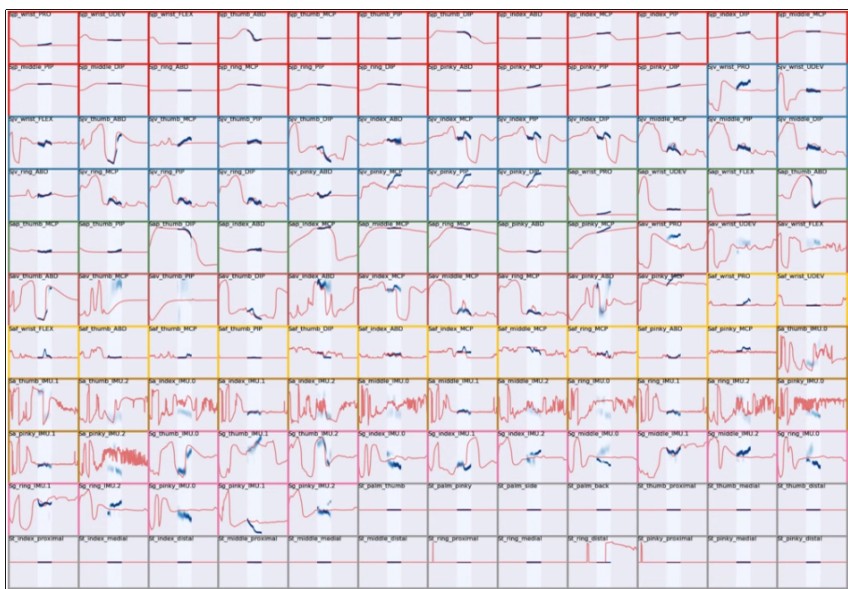

Figure 10: A visualization of the model planning to minimize predicted entropy.

| | Passive | Active | Shadow |
|---|---|---|---|
| `control_embed_depth` | 1 | 1 | 0 |
| `control_embed_hidden_size` | 128 | 128 | 48 |
| `control_embed_size` | 128 | 128 | 48 |
| `sensor_embed_depth` | 1 | 1 | 2 |
| `sensor_embed_hidden_size` | 128 | 128 | 31 |
| `sensor_embed_size` | 128 | 128 | 31 |
| `preco_hidden_size` | 128 | 128 | 34 |
| `mean_depth` | 1 | 1 | 2 |
| `mean_hidden_size` | 128 | 128 | 52 |
| `stddev_depth` | 1 | 1 | 2 |
| `stddev_hidden_size` | 128 | 128 | 122 |
| `likelihood_mixture_depth` | 1 | 1 | 2 |
| `likelihood_mixture_hidden_size` | 128 | 128 | 60 |
| `likelihood_num_components` | 2 | 2 | 2 |
| `adam_learning_rate` | 0.00025 | 0.000436490726736 | 0.00195542112406 |
| `num_overshoot_steps` | 30 | 30 | 20 |

Table 1: Hyperparameters for various dynamics models used in the experiments.

sensors that cover different regions. In total these sensors give a 132 dimensional proprioceptive state.

## D  HYPERPARAMETERS

Table 1 shows hyperparameters for several of the models used in the experiments. Some of the hyperparameters (notably the Adam learning rates) are found through random search, so the numbers are quite particular, but the particularity should not be taken as a sign of delicacy. The meaning of each parameter is shown in Figure 11.

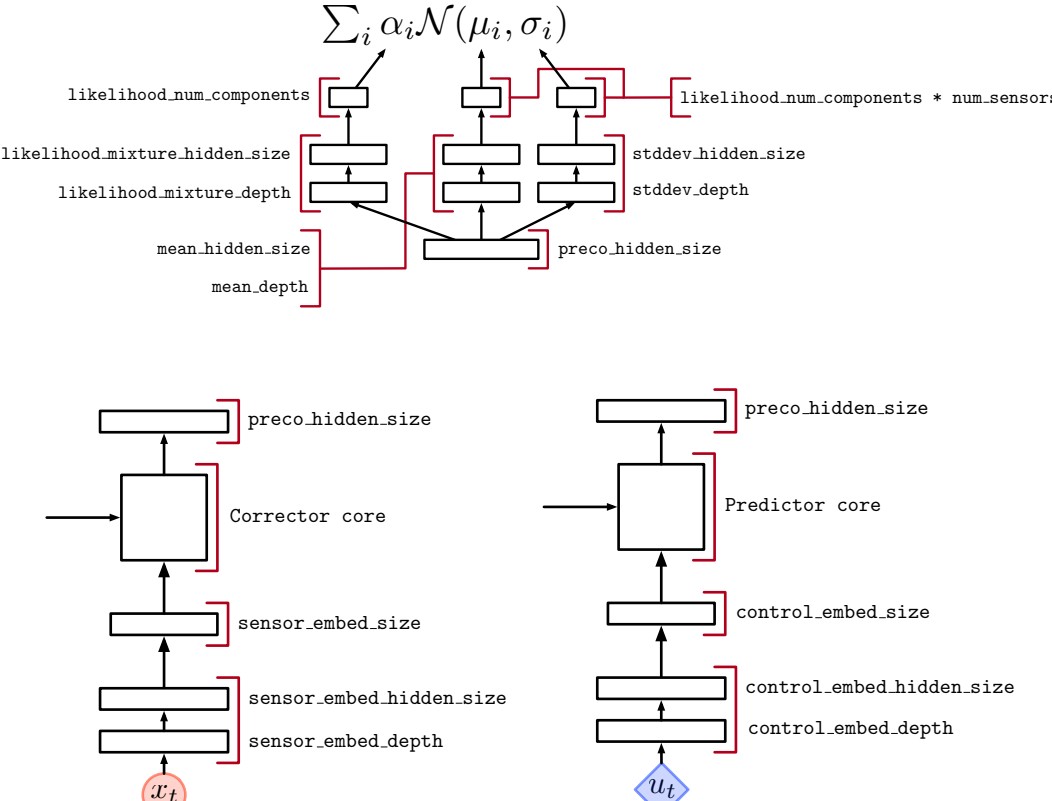

Figure 11: Detailed architecture diagrams of the components of the Preco model, along with labels that indicate different hyperparameters. A MLP sections of the models are parameterised with a depth and a hidden size, where a depth of $d$ and a hidden size of $k$ indicates $d$ hidden layers of size $k$. We do not count the output layer or the input layer in the depth parameter (so a depth of 0 is a single linear transform followed by an activation function). The output layers of the MLP parts of the model are all indicated separately in the diagrams. The three pieces shown here are attached together in various ways, as shown in Figure 3 in the main body of the paper.

