# OpenReview forum: "Learning Awareness Models"
_ICLR.cc/2018/Conference — Accept (Poster)_

### Official Review · AnonReviewer1 · 2017-11-25
**Great Paper (update: with weaknesses)**

**Rating:** 7
**Confidence:** 4

**Review:**

The paper proposes an architecture for internal model learning of a robotic system and applies it to a simulated and a real robotic hand.  The model allows making relatively long-term predictions with uncertainties. The models are used to perform model predictive control to achieve informative actions. It is shown that the hidden state of the learned models contains relevant information about the objects the hand was interacting with.

The paper reads well. The method is sufficiently well explained and the results are presented in an illustrative and informative way.
update: See critique in my comment below.
I have a few minor points:

- Sec 2: you may consider to cite the work on maximising predictive information as intrinsic motivation:
G. Martius, R. Der, and N. Ay. Information driven self-organization of complex robotic behaviors. PLoS ONE, 8(5):e63400, 2013.
- Fig 2: bottom: add labels to axis, and maybe mention that same color code as above
- Sec 4 par 3: .... intentionally not autoregressive: w.r.t. to what? to the observations?
- Sec 7.1: how is the optimization for the MPC performed? Which algorithm did you use and long does the optimization take?
 in first Eq: should f not be sampled from GMMpdf, so replace = with \sim

Typos:
- Sec1 par2: This pattern has has ...
- Sec 2 par2: statistics ofthe
- Sec 4 line2: prefix of an episode , where  (space before ,)

---

> ### Author Response · Authors · 2017-12-22
> **Thank you for the positive review**
>
> "- Sec 2: you may consider to cite the work on maximising predictive information as intrinsic motivation:
> G. Martius, R. Der, and N. Ay. Information driven self-organization of complex robotic behaviors. PLoS ONE, 8(5):e63400, 2013."
>
> Thank you for the relevant reference.
>
> "- Sec 4 par 3: .... intentionally not autoregressive: w.r.t. to what? to the observations? "
>
> Intentionally not autoregressive with respect to time.  We have clarified this in the paper.
>
> "- Sec 7.1: how is the optimization for the MPC performed? Which algorithm did you use and long does the optimization take?"
>
> We take the very naive approach of optimizing the MPC objective for a fixed number of steps by differentiating the cost function with respect to the actions and taking (projected) gradient steps with Adam. We project the actions into the constraint set at each step.  We initialize the nominal action sequence with a burn in of 1000 Adam steps (which is fairly time consuming) and we take 10 additional optimization steps after executing each action and observing a response, warm started from the previous solution.  This is acceptably fast for experimentation (2-5 steps/second after burn in) but is substantially slower than real time, primarily due to the cost of evaluating the model.
>
>
> " in first Eq: should f not be sampled from GMMpdf, so replace = with \sim"
>
> f is the Gaussian PDF defined by the parameters, not a sample from the PDF. We don't need to sample from the predictive distribution in order to compute the control objective.  One of the reasons we chose the Renyi entropy as the objective is that it is easy to compute analytically for the Mixture of Gaussians predictions that we make at each step.

---

> > ### Comment · AnonReviewer1 · 2018-01-12
> > **Revised version not as good as it could**
> >
> > I was somehow overly enthusiastic when giving my initial score because I really like the research direction and also found the technique interesting of using the Renyi entropies etc.
> > However, when looking at the paper again, I share some of the concerns of the other reviewers, mostly related to the presentation and controls.
> >
> > I am quite surprised to see that:
> > 1) the explanation of how the MPC (as given above) did not go into paper. The authors changed the explanation of the MPC but did not write that they did it with a gradient descent using ADAM and projection etc. The paper should contain enough information to reproduce it!
> > 2) none of the suggested literature  (also by the other reviewers) made it into the paper
> > 3) The authors did no attempt to improve their presentation, e.g. of Figure 4, which is indeed far from ideal (as noted by the other reviewers). Why not picking a few sensors with qualitatively different predicted certainty and show those enlarged. It would also be interesting to see what is the overall prediction performance of the model.
> > 4) no simple control experiment was done after the critic, e.g. using some random policy/grasps and compare how much more informative the "aware" actions are.
> >
> > I still think it is an important contribution. I would nevertheless urge the authors to fix as many of the above-mentioned problems as possible in the camera-ready version.
> >
> > Details:
> > 7.1 last sentence: enforce _slew_ contraints

---

### Official Review · AnonReviewer2 · 2017-11-28
**Learning a world representation through state prediction, implemented and evaluated in an unclear manner.**

**Rating:** 4
**Confidence:** 4

**Review:**

Summary:
The paper describes a system which creates an internal representation of the scene given observations, being this internal representation advantageous over raw sensory input for object classification and control. The internal representation comes from a recurrent network (more specifically, a sequence2sequence net) trained to maximize the likelihood of the observations from training

Positive aspects:
The authors suggest an interesting hypothesis: an internal representation of the world which is useful for control could be obtained just by forcing the agent to be able to predict the outcome of its actions in the world. This hypothesis would enable robots to train it in a self-supervised manner, which would be extremely valuable.

Negative aspects:
Although the premise of the paper is interesting, its execution is not ideal. The formulation of the problem is unclear and difficult to follow, with a number of important terms left undefined. Moreover, the experiment task is too simplistic; from the results, it's not clear whether the representation is anything more than trivial accumulation of sensory input

- Lack of clarity:
-- what exactly is the "generic cost" C in section 7.1?
-- why are both f and z parameters of C? f is directly a function of z. Given that the form of C is not explained, seems like f could be directly computing as part of C.
-- what is the relation between actions a in section 7.1 and u in section 4?
-- How is the minimization problem of u_{1:T} solved?
-- Are the authors sure that they perform gathering information through "maximizing uncertainty" (section 7.1)? This sounds profoundly counterintuitive. Maximizing the uncertainty in the world state should result in minimum information about the worlds state. I would assume this is a serious typo, but cannot confirm given that the relation between the minimize cost C and the Renyi entropy H is not explicitely stated.
-- When the authors state that "The learner trains the model by maximum likelihood" in section 7.1, do they refer to the prediction model or the control model? It would seem that it is the control model, but the objective being "the same as in section 6" points in the opposite direction
-- What is the method for classifying and/or regressing given the features and internal representation? This is important because, if the method was a recurrent net with memory, the differences between the two representations would probably be minimal.

- Simplistic experimental task:
My main intake from the experiments is that having a recurrent network processing the sensory input provides some "memory" to the system which reduces uncertainty when sensory data is ambiguous. This is visible from the fact that the performance from both systems is comparable at the beginning, but degrades for sensory input when the hand is open. This could be achievable in many simple ways, like modeling the classification/regression problem directly with an LSTM for example. Simpler modalities of providing a memory to the system should be used as a baseline.


Conclusion:
Although the idea of learning an internal representation of the world by being able to predict its state from observations is interesting, the presented paper is a) too simplistic in its experimental evaluation and b) too unclear about its implementation. Consequently, I believe the authors should improve these aspects before the article is valuable to the community

---

> ### Author Response · Authors · 2017-12-22
> **Thank you for the detailed feedback**
>
> "Although the premise of the paper is interesting, its execution is not ideal. ... it's not clear whether the representation is anything more than trivial accumulation of sensory input"
>
> We hope the general reply and videos help with this.
>
> "what exactly is the "generic cost" C in section 7.1?"
> "why are both f and z parameters of C? f is directly a function of z. Given that the form of C is not explained, seems like f could be directly computing as part of C."
>
> In the MPC presentation in section 7.1 we intentionally left the objective generically as C and agree that our current presentation is confusing. For example, to maximize the entropy in the nominal trajectory for exploration, the objective C is the (negated) Renyi entropy of the model predictions, which directly uses the GMM PDF f.
>
> You are correct that the PDF f and hidden state z need not necessarily be parameters of C. However in some cases it adds notational convenience; the PDF f is useful for when the goal of the objective is to reach a desired state (such as the maximum entropy, or to maximize the fingertip pressure), and the hidden state z is useful for when the goal of the objective extracts other information from the hidden state (such as the type of object, or the height of the object)
>
> We have clarified this portion in the paper.
>
> "what is the relation between actions a in section 7.1 and u in section 4?"
>
> Minimizing the objective in 7.1 over a obtains u.  We agree this is probably not the best notation and we've updated the description in 7.1 to use u and u^star instead.
>
> "How is the minimization problem of u_{1:T} solved?"
>
> This problem is solved using Adam with warm starts from the previous step, see our response to R1 for a full description.
>
> "Are the authors sure that they perform gathering information through "maximizing uncertainty"..."
>
> We perform information gathering through maximizing (and not minimizing) uncertainty.  To understand why this is the case it is important to be clear about exactly which uncertainty is being maximized, and what we are trying to gather information about.
>
> It is true that in some settings (e.g. A Bayesian exploration-exploitation approach for optimal online sensing and planning with a visually guided mobile robot by R Martinez-Cantin, N de Freitas, E Brochu, J Castellanos, and  A Doucet in Autonomous Robots 27 (2), 93-103, and the many references therein) one might choose to minimize uncertainty of the internal belief state, so as to gather information. This also arises naturally in Bayesian experimental design (eg the highly cited work of K Chaloner).
>
> However, here we are concerned with the uncertainty in the predictions of the hand signals (touch sensors, vestibular info, and proprioception). By seeking to maximize this uncertainty, the hand is driven to try behaviours where it is highly uncertain about what sensations it will experience (eg what its pressure sensors will feel).
>
> To further emphasize this consistency, we show the outcome of instead acting to minimize the predicted entropy in Video 8 (Figure 3(c) in the paper).  Under this objective the hand pulls itself away from the block.
>
> "When the authors state that "The learner trains the model by maximum likelihood" in section 7.1, do they refer to the prediction model or the control model?"
>
> The only model in this paper is the predictive model.  Control is implemented as planning in the predictive model using MPC.  The learner trains the predictive model, and the actors plan trajectories using an objective that depends on the predictive model, but there is no separate learned control model involved.
>
> "What is the method for classifying and/or regressing given the features and internal representation?"
>
> The diagnostic models are MLPs that look at a single time step of the predictive model state. Your supposition is correct that if we use a recurrent net as the diagnostic model then there is no advantage to including the predictive model features.  It would be quite surprising if this were not the case.
>
> The purpose of the diagnostic model ties back to information partitioning.  The role of the diagnostic is to show that although we do not have an explicit representation of any external state in the predictive model, we nonetheless obtain such a representation implicitly (since if this were not the case the diagnostic task would fail).
>
> "Simplistic experimental task:..."
>
> The motivation behind our setup is that collecting the data to train the "direct" model can be difficult or impossible in a real life setting.  The most complex part of the apparatus for the shadow hand experiment (apart from the hand itself) is the mechanism for measuring the angle of the object being grasped.

---

### Official Review · AnonReviewer4 · 2017-12-12
**Some really good ideas, but probably not ready for publication**

**Rating:** 4
**Confidence:** 5

**Review:**

The authors explore how sequence models that look at proprioceptive signals from a simulated or real-world robotic hand can be used to decode properties of objects (which are not directly observed), or produce entropy maximizing or minimizing motions.

The overall idea presented in the paper is quite nice: proprioception-based models that inject actions and encoder/pressure observations can be used to measure physical properties of objects that are not directly observed, and can also be used to create information gathering (or avoiding) behaviors. There is some related work that the authors do not cite that is highly relevant here. A few in particular come to mind:

Yu, Tan, Liu, Turk. Preparing for the Unknown: uses a sequence model to estimate physical properties of a robot (rather than unobserved objects)

Fu, Levine, Abbeel. One-Shot Learning of Manipulation Skills: trains a similar proprioception-only model and uses it for object manipulation, similar idea that object properties can be induced from proprioception

But in general the citations to relevant robotic manipulation work are pretty sparse.

The biggest issue with the paper though is with the results. There are no comparisons or reasonable baselines of any kind, and the reported results are a bit hard to judge. As far as I can understand, there are no quantitative results in simulation at all, and the real-world results are not good, indicating something like 15 degrees of error in predicting the pose of a single object. That doesn't seem especially good, though it's also very hard to tell without a baseline.

Overall, this seems like a good workshop paper, but probably substantial additional experimental work is needed in order to evaluate the practical usefulness of this method. I would however strongly encourage the authors to pursue this research further: it seems very promising, and I think that, with more rigorous evaluation and comparisons, it could be quite a nice paper!

One point about style: I found the somewhat lofty claims in the introduction a bit off-putting. It's great to discuss the greater "vision" behind the work, but this paper suffers from a bit too much high-level vision and not enough effort put into explaining what the method actually does.

---

> ### Author Response · Authors · 2017-12-22
> **Thank you for your expert feedback.**
>
> "Yu, Tan, Liu, Turk. Preparing for the Unknown..."
> "Fu, Levine, Abbeel. One-Shot Learning of Manipulation Skills..."
>
> Thank you for these very relevant references. There are similarities, but important differences. The paper of Yu and colleagues uses labels of the world properties (mu in their notation) to pre-learn the model in a supervised way. We on the other hand aim to show that properties of the world come to be represented in a model that is only trained with body labels. While relevant the two papers are markedly different. The paper of Lu and colleagues, and in fact may of their subsequent works including guided policy search, use classical control ideas and iterate between fitting trajectories and improving control policies. We have shown that we can learn more complex models and exploration policies jointly. The works are related, but with many differences, and we feel it will be promising to explore this connection further. Thank you for pointing out this connection.
>
> "But in general the citations to relevant robotic manipulation work are pretty sparse."
>
> We agree. We will address this.
>
> "The biggest issue with the paper though is with the results."
>
> To demonstrate what we are referring to as “awareness” in this paper, we do include baselines for all of our experiments that perform the same tasks using the raw sensor state instead of the hidden state. Please see also the general response and videos, which include plots and comparisons for reference.
>
> In the pose-prediction experiment on the Shadow hand, our baseline model (in Figure 5) is called “sensors” and tries to predict the pose of the object from only sensory information (without the hidden state). This baseline also does not surpass the ~15 degrees of error, indicating that the sensor readings only contain coarse-grained information about the object and that anything better than ~15 degrees is not possible with the current setup. Our approach outperforms this baseline in most cases and shows that the hidden state of the predictive models have learned a useful representation. We will revise our paper and make the baselines more clearly labeled.
>
> "I found the somewhat lofty claims in the introduction a bit off-putting."
>
> We agree.  We've modified the intro to remove some of the more lofty claims.

---

> > ### Comment · AnonReviewer4 · 2017-12-23
> > **Response**
> >
> > I appreciate the additional details provided in the response, and the additional videos. However, I don't think that the authors have really addressed my main concern: the difficulty of judging the quality of the results. The additional results in the videos are purely qualitative, and it's impossible to tell what works and what doesn't. In the paper, there are three results figures: Figure 4 is essentially impossible to understand, it's a wall seemingly random plots. Figure 2 appears to show that, on some time steps, directly regressing from sensors does as well as the proposed model, but the proposed model tends to do well on more time steps. This is not surprising -- if the sensors at the "best" time step are sufficient to infer the object, then a recurrent model that tries to track and predict future sensory readings will probably have hidden state that correlates with history, making it slightly better for inferring the information about the objects. How hard or easy this task is, or how significant this result is, is impossible to determine here. In the real world results, I'm not convinced by the 15 degree error. There is again no serious baseline, and the lower bound baseline (a feedforward memoryless network) seems to do about equally well most of the time. Simply using a recurrent model would provide at least an upper bound baseline, but realistically, if this task is really too hard (or too easy?), perhaps it's just not the right task for evaluating the method.
> >
> > There are some additional model-based control results provided in the videos. They look very interesting and promising. But these too are hard to interpret without a serious quantitative evaluation. It seems that sometimes the hand figures out that closing fingers correlates with higher force -- that makes sense, and it suggests that model-based RL works. There are other recent papers on model-based RL too, including using neural network models (including many papers you don't cite).
> >
> > Regarding the two prior works: it doesn't seem like they are cited in the updated draft. I understand these papers are not the same, but I disagree with the authors that they are irrelevant to cite. They aren't. Specifically:
> >
> > Yu et al. should really be compared to. In fact, I think you might actually be doing this in the videos (the LSTM baseline), but since this information is not in the paper (and neither is the citation), it's impossible for me to tell. Yes, Yu et al. is doing something different, and I get that your paper uses "less information" in some sense. I would argue it doesn't, it's just a peculiarly circuitous way to learn to identify objects, but I do think your viewpoint on this is reasonable, if debatable. However, there still needs to be an honest effort at a comparative evaluation and proper overview of the relevant literature.
> >
> > Fu et al. (which I assume is what you call "Lu et al") should be cited and discussed, as should other recent model-based RL work. You say "there have been many success stories of model-free RL with deep neural representations, the same cannot be said about model-based RL" -- well, here is a paper doing model-based RL that predates yours, and while it is doing different tasks, clearly the idea is related. There are other papers that do model-based RL too, many of which are not cited or compared to. Simply asserting that they are not "successful" is not going to cut it, the burden is on you to compare and/or discuss, instead of pretending that they don't exist.
> >
> > To be clear: I do think that there is something really interesting about this paper. I think what that something is is an incremental but valuable improvement in training predictive models for model-based RL. I would really like to give it a higher score, but in its present state, this just doesn't seem like a paper I would expect to see published in an academic conference, but rather a workshop paper. The rigorous evaluation, comparison to prior work, and a serious attempt to summarize prior work in this area is missing. I think this is unfortunate, because I do suspect things actually work reasonably well in some cases, and if the authors scope their claims properly, acknowledge what works and what doesn't, actually provide rigorous quantitative evaluations, and properly position their paper in regard to prior work, I think it would be a good paper.

---

### Author Response · Authors · 2017-12-22
**General response from the authors**

We thank the three reviewers. Their comments are not only valid but also very helpful. They will no doubt help us improve the presentation of this work considerably.

We will address the reviewers’ questions in detail individually. To address the general concern about baselines and results, we have experimented with several variants of our model. While time prohibits us from offering full quantitative ablations at this stage, we have produced some videos (included as a comment in this thread) to provide a more clear illustration of what we have accomplished using one of our models. (We agree that in the paper we could do a lot better to communicate our results.)

In summary, we have shown that while learning to explore by maximising Renyi entropy of body predictions, it is possible to simultaneously learn dynamic, long-term, predictive neural network models of body measurements, including proprioceptive signals, vestibular information and touch sensors. We have shown that the learned body models can be used to solve other control tasks. While there have been many success stories of model-free RL with deep neural representations, the same cannot be said about model-based RL. In this sense, this paper puts forward a rare example of learning to control while (simultaneously) learning neural dynamic models effectively. It however goes beyond this to provide evidence of the value of embodiment in AI, in particular that it is possible to learn persistent, holistic, dynamic representations of the world by simply learning to predict body signals. We believe this position could be helpful to advance research in AI.

---

> ### Author Response · Authors · 2017-12-22
> **Videos**
>
> Videos 1 (https://www.youtube.com/watch?v=BogIU66kfpo) and 2 (https://www.youtube.com/watch?v=jyAJEcFybpI) show how we can successfully predict (blue) 132 hand measurements (red) up to 500 steps ahead! The learned online, probabilistic predictive models are very effective at long-term prediction, and not just predicting the next observation.
>
> In Video 3 (https://www.youtube.com/watch?v=fCm7iQdFXCs) we show the results of regressing from the internal states of the hand model --- which was only trained to predict the 132 hand signals, ie proprioception, vestibular info and touch sensors --- to the shape of the object that was touched (in this video, our model is called PreCoNN). It is crucial to emphasize that what we are aiming to demonstrate here is unsupervised learning, not supervised learning. That is, we train the body model with body labels (proprioception, touch and vestibular info) only. We then want to show that the learned representations of the body model can be used to predict properties of objects in the world. That is by modelling the body only, we want to show that we can acquire persistent knowledge about the shape of objects in the world. The motivation for this being that the world is very complicated and ever changing, so it might be easier to focus on modelling the body.
>
> We argue that the learned representations are persistent because even when the hand is not touching the object, the model is still aware of the shape of the object. To show that this is true, that is that information about object shape is present in the body model, we use a simple MLP (crucially with not time dependence) to predict shape. We only do this as a diagnostic and not because this is the final goal. Again the end goal is not to predict shape from touch, but rather to show that a model trained only on body measurements can encode information about shapes of objects in the world from recent interactions. We simply want to show this information is there.
>
> If we had object shape labels, we could simply train supervised models (eg feedforward NNs or LSTMs). If we do this we get the other videos (RawNN and RawLSTM) as shown in Video 3. Clearly LSTM which is trained in a supervised manner does better. NN doesn’t model dynamics and so it struggles when the hand stops touching the object. However, what is cool about our result is that PreCoNN was pre-trained without access to object shape labels. There was a initial phase of unsupervised learning to obtain all the features, these are then fixed and subsequently a simple MLP is sufficient to predict the shape of the object from the state of the forward model. That is by modelling the body only, we can learn good features for doing predictions about the world.
>
> In Video 4 (https://www.youtube.com/watch?v=qiez5Bziyp8), we show the behaviour of the hand using the training object of maximizing the Renyi entropy. Note the hand seeks to touch the objects and manipulate them!
>
> Videos 5-8 show that the learned model can be used with novel control objectives, respectively maximizing the Renyi entropy of the fingertips only (https://www.youtube.com/watch?v=3Hh5WQ5HeSs) with the fingertips learning to touch each other, maximising pressure on the fingertips (https://www.youtube.com/watch?v=TCiQZfpR1zc) with the fingertips seeking contact, maximizing pressure for the fingers and palm sensors (https://www.youtube.com/watch?v=mbBpfWDA6B4), and minimizing Renyi entropy for all sensors (https://www.youtube.com/watch?v=bHmzG6mxTh0). In the last video, the hand avoids touch the object as one would expect because in this case it has no uncertainty in predicting what it feels. All these results are consistent with each other.
>
> Finally, Video 9 (https://www.youtube.com/watch?v=ojTSIb6OT9w) shows the data collection part of our Shadow hand experiment.
>
> All videos collected into a single playlist: https://www.youtube.com/playlist?list=PLwtvFFFFBVYL5GzMxq40_fjNUQD91vw4O

---

### Decision · Program_Chairs · 2018-01-29
**ICLR 2018 Conference Acceptance Decision**

**Decision:**

Accept (Poster)

**Comment:**

Since this seems interesting, I suggest to accept this paper at the conference. However, there are still some serious issues with the paper, including missing references.